# Investigation of Radiation Safety Pictogram Recognition in Daily Life

**DOI:** 10.3390/ijerph18042166

**Published:** 2021-02-23

**Authors:** Kyoungho Choi, Jinhee Choi

**Affiliations:** 1Radiological Science/Research Institute of Health Statistics, Jeonju University, Jeonju-si 55069, Korea; ckh414@jj.ac.kr; 2Department of Fashion Business, Jeonju University, Jeonju-si 55069, Korea

**Keywords:** radiation safety, pictograms, natural radiation, medical radiation, mean comparison

## Abstract

After the Fukushima Daiichi nuclear accident of 2011, interest regarding radiation safety in everyday life has increased considerably. This study investigates the general public’s current level of awareness of six warning pictograms in regard to medical and natural radiation safety utilized under ISO 7010, as per Korea industrial regulations. Namely, it tests whether survey respondents can recognize pictograms related to radiation safety according to their purpose, as their inability to do so poses a serious safety problem. The empirical analysis results regarding the awareness levels for radiation safety pictograms are as follows. First, 63.3% of the respondents were unable to correctly identify the pictograms; that is, their level of understandings of the six pictograms related to everyday radiation were low. Second, the mean score for the correct responses to the question of what the six pictograms indicated in relation to everyday radiation safety was also relatively low, with a mean score of 2.79 and a standard deviation of 1.447. The primary reasons for the low awareness and understanding levels were identified to be insufficient education related to radiation safety in schools. Additionally, it is necessary to revise and rectify current warning pictograms established by the Korea Industrial Standards and ISO 7010. This study is thus significant in that it identifies the level of understanding of the pictograms and suggests the need for improvement as a diversified effort toward improving everyday radiation safety.

## 1. Introduction

We are unknowingly exposed to radiation over the course of our daily lives. In general, radiation includes natural radiation from all materials on Earth and artificial radiation that humans generate for a particular purpose. Generally, artificial radiation is regulated by the Nuclear Safety Act and regulations enacted under the guidelines of the International Atomic Energy Agency (IAEA) [1]. Furthermore, natural radiation is widespread in our everyday lives and is typically present in our proximity, meaning people are excessively exposed to unnecessary radiation in their daily lives, which poses significant risks to their health.

The management and regulation of natural radiation can be considered meaningless, but much of the radiation the general public is exposed to is natural. Notable examples include household goods, such as the stone plates of some thermal beds, health bracelets or necklaces, wallpapers, pillows [2], or long-term stays in subway stations with high radon concentrations. The Radiation Safety Law was thus enacted on 25 July 2011 to protect the people and environment from everyday radiation exposure [3]. Nevertheless, safety management of natural radioactive materials and of everyday radiation exposure is insufficient. Accordingly, this led to studies such as that of Yoon [4], who recognized the danger of harmful chemicals, including the radioactive material contained in everyday items, and highlighted the need for prevention.

However, there seems to be no significant change regarding the safety management of radiation emissions in everyday life. Originally, safety referred to a “comfortable state where humans are at ease with no anxiety over the possibility of bad results on their welfare without any intimidation of their existence” [5]. However, since the Sewol Ferry disaster, South Korea has increased its awareness for safety, and in 2016, it proposed a pan-governmental life cycle safety education plan customized to all ages, which presents the education requirements necessary for personal safety. The proposal included a plan to promote life cycle safety education and seven major standards for school safety education, comprised of medium classifications and 56 minor classifications [6]. Nevertheless, although safety-related risks have increased in everyday life along with the overall quality of life, a surge in safety insensitivity within society and the need for safety is relatively widespread [7]. However, there is no information on radiation safety in everyday life for the above-mentioned safety education standard.

Meanwhile, to ensure the safety of the general public, 300 pictograms for public facilities such as toilets, restaurants, and subways have already been established and used as the industrial standard in Korea (KS). According to Huh [8], a public information pictogram is defined as a “picture signal composed to deliver a message to public.” These pictograms are designated and used as international standards in many countries for providing guidance in public places or public facilities. However, pictograms should be interpreted without visual misunderstanding of their purpose to be able to provide warnings, guidance, and protection. They should also be independent of cultural norms and based on international standards. Accordingly, the International Organization for Standardization (ISO) has standardized the different pictograms, establishing ISO 7001 (public information symbols) for information pictograms in public facilities and ISO 7010 (standard for safety symbols) for safety information pictures.

Based on this, the Korea Standards Service Network proposed six categories of pictograms related to facilities and five categories related to safety (forced action, warning, prohibition, emergency or evacuation, and firefighting equipment). Pictograms are thus used as effective complementary and alternative communication methods not only for non-disabled people, but also for people with communication disabilities [9]; they are, therefore, highly valuable. There are studies on safety-related pictograms being used as a communication method, including that of Caffaro and Cavallo [10] in the field of agricultural machinery and those of Clayton and Perlotto [11] and Jentsch [12] in the field of aviation, but there is little research on radiation safety-related pictograms.

If the general public does not fully understand and recognize the meanings of pictograms related to safety, serious problems may sometimes arise, potentially posing risks to the health of the general public. This is especially true for warning pictograms concerning medical and natural radiation safety. At a time during which radiation safety is required in everyday life, the nature and use of radiation safety-related pictograms differ from those used for public facilities and events. Therefore, regulatory management is required, and more emphasis should be placed on accurate semantic transfer than on originality [13].

It is important to investigate the level of awareness, how accurately the general public understands safety-related pictograms, and find solutions. In this study, we examine the awareness of six pictograms related to radiation safety from among the warning pictograms presented in ISO 7010, derive areas for improvement through statistical analysis, and propose the need for radiation safety education. To this end, we try to answer the following research questions: (1) What is the level of understanding and rate of correct answers for the six presented pictograms? (2) Are there statistically significant differences among genders and ages regarding the rate of correct answers? (3) Is there a statistically significant correlation between the scores, levels of understanding, and adequacy of the six presented pictograms. Lastly, (4) what is the statistical model that can predict the adequacy using the level of understanding? Finally, we present measures that can increase awareness for pictograms related to radiation safety.

## 2. Methods

### 2.1. Outline of Pictograms

A pictogram is a graphic symbol which refers to the meaning of a specific concept as a means for removing language barriers [14]. Specifically, picto refers to a picture, and gram refers to a message (as in telegram). As such, pictograms have been rapidly disseminated and developed as a consistent means of communication in modern societies, in which information needs to be delivered faster and more accurately. Therefore, the use of pictograms is constantly increasing [15].

There are three functions of pictograms: guidance (notice), command, and symbol. The guidance function is primarily represented by arrows that provide direction, direction indicators on road signs, and guidance toward facilities. The command function leads to action and is used for various road signs, prohibition signs, and danger signs, generally demonstrating rules and actions that people must follow. The symbol function includes quality-related indicators, symbols such as the addition and subtraction symbols in mathematics, musical notes, and geographical symbols indicating the state or characteristics of land [16]. The pictograms referred to in this study represent radiation safety-related warnings and correspond to the command function. There are prior studies on the various functions of pictograms in several areas, including Olympic pictogram recognition [17] and its trends [18], Expo pictogram design recognition [14], familiarity of pictograms related to facilities [19], and safety pictogram recognition in everyday life [20], among others.

### 2.2. Study Subject and Materials

Using a prior study [14] that examined the recognition of pictograms for reference, the subjects of this study were people from various age groups, ranging from those in their 20s to those in their 50s or older. The process of selecting respondents was not only aimed at those engaged in certain occupations, such as the health and medical fields, as we aimed to identify the general public’s awareness of radiation safety-related pictograms, including those with no professional knowledge. The survey was conducted from January 2 to March 31, 2020. We used convenience sampling to select 200 respondents residing in Jeollabuk-do, mainly in Jeonju City. Data were collected through the self-indication method [21].

### 2.3. Survey Composition

The survey consisted of two main areas. The first was focused on the recognition of the radiation-related pictograms in Figure 1, and the second referred to demographic details. The second section consisted of questions regarding gender and age, how easy it was to understand radiation-related pictograms, their adequacy, and any prior knowledge of radiation.

(Question) The following figure is an internationally designated warning picture (pictogram) regarding radiation safety. Please indicate each picture conveys by matching each picture with right-hand side of the image.

### 2.4. Ethical Considerations and Tools for Statistical Analysis

We informed the respondents of the purpose of the study, announcing to them that confidentiality was guaranteed, that they could stop answering at any given moment without adverse consequences, and that they had the right to discontinue the study. We obtained their consent before collecting the data. The survey took around five minutes, and the respondents were paid a small monetary amount as a token of appreciation. Subsequently, statistical analysis was conducted using an IBM SPSS25 on the collected data entered into Excel, and the insincere responses were removed during preprocessing [19]. Regarding demographics, the respondent characteristics were identified using descriptive statistical analysis, while for the radiation-related pictogram recognition section, an independent sample t-test, analysis of variance, and correlation and regression analysis were utilized.

## 3. Statistical Analysis Results

### 3.1. Descriptive Statistics

As is shown in Table 1, out of 199 respondents, 74 were male (37.2%) and 125 were female (62.8). Classified by age group, the largest group of respondents (75 respondents, or 37.7%) was under 29, followed by those in their 30s (24.6%), those in their 50s (23.1%), and those in their 40s (14.6%). Regarding the recognition of the six pictograms related to everyday radiation shown in Figure 1, 63.3% of the respondents did not identify any pictograms, and only 6.0% identified all of them. We thus concluded that it was difficult for the general public to understand the pictograms related to everyday radiation currently adopted and used as per Korean industrial standards.

### 3.2. Evaluation of the Level of Understanding and the Rate of Correct Answers

The mean response to the question regarding the meanings of the six pictograms in Figure 1 was 2.79, as shown in Table 2, and the standard deviation was 1.447. As for the rate of correct answers by pictogram, as shown in Table 3, the magnetic field pictogram (P6) had the highest rate at 87.9%, followed by the radiation material pictogram (P1) at 67.3% and the non-ionizing radiation pictogram (P3) at 16.1%. The reason for this high level of understanding for the pictograms of magnetic fields, radiation, and materials is that the magnetic field pictogram takes the form of a magnet, which is a relatively easy concept to understand, and the radioactive material and ionizing radiation pictograms are relatively easy to observe in medical institutions.

Independent t-testing to determine whether there were statistically significant differences in the scores of the correct answers for each gender showed that the difference in the scores for the male and female respondents was statistically significant at a significance level of 5%, as shown in Table 4, in the absence of Levene’s equal variance (*p* = 0.044). In other words, male respondents’ correct answers were statistically significantly higher compared with those of the female respondents. These results are, in part, similar to Park’s [16] study of Olympic pictogram awareness. An explanation as to why there was such a significant difference in recognition by gender may arise from the different brain structure characteristics for each gender; that is, males tend to have a more developed left brain hemisphere, which is responsible for spatial sensation, including sequences, logic, and mathematics, making them more aware of warning signs and pictograms than females, who have a more developed right brain hemisphere, which is responsible for sensitivity.

A one-way analysis of variance was then conducted to determine whether there were statistically significant differences in the scores of the correct answers by age, and the results are shown in Table 5. Duncan’s post hoc analysis results showed differences between those in their 20s and those in their 50s and older.

### 3.3. Correlation and Regression Analysis

Furthermore, correlation analysis was performed to determine whether there was a statistically significant correlation between the scores for the question about the meanings of the six pictograms in Figure 1 and the level of understanding and adequacy of these pictograms. As shown in Table 6, there was a statistically significant relationship between the scores and adequacy at a significance level of 5%. In other words, the respondents who were well aware of the meanings of the pictograms were able to make appropriate judgments. Furthermore, the correlation coefficient between the level of understanding and adequacy was relatively high (0.577), which shows that the higher the level of understanding was, the higher the adequacy.

To establish a statistical model that could predict the adequacy by using the level of understanding, a simple linear regression analysis was conducted, and the results are shown in Table 7. With a *p*-value of 0.000 (<0.05), the model was statistically significant at a significance level of 5%, and its explanatory power was 0.333.

## 4. Conclusions

Radiation has been present in nature and has coexisted with humanity long before humans started using it in a peaceful manner. In addition to being exposed to medical radiation during the course of treatment, individuals are also exposed to natural radiation from soil, air, food, and the cosmos in their daily lives. As such, radiation is one of the energy sources we always encounter in our living environment. Exposure to such radiation can have biological effects, such as changes, damage, and even harm, depending on the extent of the exposure. As a result, in Korea, the Nuclear Safety Law or Radiation Safety Control Law on everyday radiation have been enacted to help ensure our health and safety from radiation. As the general public is not familiar with radiation, warning signs and pictures have been developed and utilized to help ensure their safety. To find out how easily these pictograms are recognized by the general public, this study conducted an empirical analysis on the understanding level of pictograms related to everyday radiation safety, as established and used in Korean industry standards. The results were as follows.

First, 63.3% of the respondents responded negatively regarding their level of understanding of the six pictograms related to everyday radiation, and only 6.0% responded positively. Furthermore, in terms of pictogram adequacy, the negative responses (55.8%) were almost five times higher than the positive ones (11.6%). Regarding their knowledge related to radiation safety in everyday life, only 42.2% chose radon (Rn) as the answer. We therefore concluded that the level of knowledge regarding the risks of radon gas is low.

Second, only 42.2% of the respondents chose radon as the correct answer when inquired about their knowledge of radiation safety in their lives. As such, respondents’ knowledge about the risk of radon gas was thus deemed to be low.

Third, for the question of what the pictograms related to everyday radiation safety meant, the mean response score was relatively low at 2.79 (standard deviation of 1.447). Furthermore, the rate of correctly matching the pictograms with their descriptions was relatively high for the pictograms that were relatively easy to guess or were widely used. However, for the pictograms on laser beams, biological hazards, and non-ionizing radiation, the correct response rates were lower. These results were due to the lack of formal education (schooling) and guidance related to radiation safety. This is also demonstrated by the fact that the seven standards for safety education in South Korean schools, implemented in 2016, do not contain any information on radiation safety.

Fourth, for the mean comparisons among genders regarding the rate of correct answers, male respondents’ scores were statistically significantly higher than those of female respondents at a significance level of 5%. The correlation coefficient between the level of understanding and the adequacy of the presented pictogram was relatively high, being 0.577. The higher the level of understanding was, the higher the adequacy.

Based on the above results, awareness of the pictograms used for radiation safety in everyday life is not high. As such, we would like to suggest ways to improve this in the future.

First, education related to everyday radiation safety should be reinforced at the school level. To do so, safety training standards must include areas related to radiation safety. Second, the current warning pictograms enacted by ISO 7010 concerning the safety of daily radiation should be modified to deliver the message more clearly. Furthermore, in the case of South Korea, warnings are depicted only as pictograms. However, if these are presented along with text, the level of understanding could be further enhanced.

This study is significant in that it identified the actual conditions related to radiation safety in everyday life and used pictograms to improve radiation safety awareness.

## Figures and Tables

**Figure 1 ijerph-18-02166-f001:**
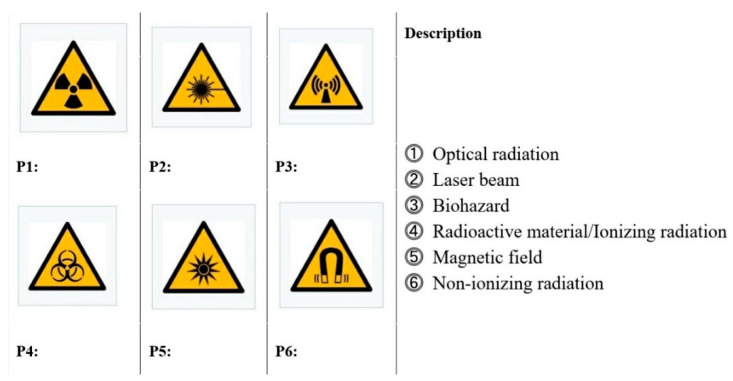
Item used to measure the recognition of pictograms related to radiation.

**Table 1 ijerph-18-02166-t001:** Demographics and the levels of comprehension and aptitudes.

Variable		Number (%)
Gender	male	74 (37.2)
	Female	125 (62.8)
Age	Under 29	75 (37.7)
	30–39	49 (24.6)
	40–49	29 (14.6)
	Over 50	46 (23.1)
Understanding	Not at all	35 (17.6)
	Limited	91 (45.7)
	Average	61 (30.7)
	Yes	12 (6.0)
Adequacy	None	24 (12.1)
	Little	87 (43.7)
	Average	65 (32.7)
	Yes	20 (10.1)
	Definitely yes	3 (1.5)
Knowledge	CO_2_	40 (20.1)
	CO	47 (23.6)
	Rn	84 (42.2)
	N	16 (8.0)
	SO	12 (6.0)

**Table 2 ijerph-18-02166-t002:** Scores for the understanding level.

	N	Min.	Max.	Mean	SD
Score	199	0	6	2.79	1.447

**Table 3 ijerph-18-02166-t003:** Rate of correct answers by pictogram (%).

	P1	P2	P3	P4	P5	P6
Correct Answer Rates	67.3	51.3	16.1	30.7	29.1	87.9

**Table 4 ijerph-18-02166-t004:** Comparison of mean scores by gender.

Gender	Mean	SD	t	*p*-Value
Male	3.22	1.616	3.056	0.003 < 0.05
Female	2.54	1.280		

**Table 5 ijerph-18-02166-t005:** Comparison of mean scores by age.

Age	Group 1	Group 2	F	*p*-Value
Over 50	2.41		2.297	0.079
40–49	2.59			
30–39	2.84			
Under 29		3.07		

**Table 6 ijerph-18-02166-t006:** Pearson’s correlation coefficient (significance level = 0.05).

	Score	Understanding	Adequacy
Score	-	0.126 (0.077)	0.164 (0.021 < 0.05)
Understanding	0.126 (0.077)	-	0.577 (0.000 < 0.05)
Adequacy	0.164 (0.021 < 0.05)	0.577 (0.000 < 0.05)	-

**Table 7 ijerph-18-02166-t007:** Regression analysis.

	B	*p*-Value	R^2^
Constant	1.041	0.000	0.333
Understanding	0.627		

## Data Availability

The data presented in the study are available on request from the corresponding authors.

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
