# Peer review of "Investigation of Radiation Safety Pictogram Recognition in Daily Life"

_ijerph, 2021, doi:10.3390/ijerph18042166_

Round 1

Reviewer 1 Report

Suggested revisions and comments are included on the attached file.

Author Response

Thank you for your review, and Please see a revised manuscript in the attachment.

Comment 1: The introduction should briefly place the study in a broad context and highlight why it is important. It should define the purpose of the work and its significance. The current state of the research field should be carefully reviewed, and key publications cited. Please highlight controversial and diverging hypotheses when necessary. Finally, briefly mention the main aim of the work and highlight the principal conclusions. As far as possible, please keep the introduction comprehensible to scientists outside your field of research. References should be numbered in order of appearance and indicated by a numeral or numerals in square brackets—e.g., [1] or [2,3], or [4–6]. See the end of the document for further details on references.

Response:

We made significant revisions to the introduction (see the revised paper). A description of the international standard for radiation safety-related pictograms, ISO 7010, was added along with related international papers.

To clarify the purpose of the study, the research problem was defined as follows.

To this end, we try to answer the following research questions: (i) what is the level of understanding and rate of correct answers for the six presented pictograms; (ii) are there statistically significant differences among genders and ages regarding the rate of correct answers; (iii) is there a statistically significant correlation between the scores, level of understanding, and adequacy of the six presented pictograms; and (iv) what is the statistical model that can predict the adequacy using the level of understanding? Finally, we aim to present measures for improvement that can increase awareness for pictograms related to radiation safety.

The citations have been modified to follow the format of the journal.

Comment 2: This section may be divided by subheadings. It should provide a concise and precise description of the experimental results, their interpretation, as well as the experimental conclusions that can be drawn.

Response:

As per your comment, Section 3 was renamed Statistical Analysis Results and divided into three subsection. Section 3.1 presents the descriptive statistics of the respondents. Section 3.2 is now titled Evaluation of the level of understanding and the rate of correct answers. In addition, we compared our results with those of prior studies, in which statistically significant differences occur between men and women in the mean comparison.

Reviewer 2 Report

In this paper, the authors studied the exact recognition level of 6 warning pictograms related to medical and natural radiation safety enacted and used in accordance with the Korea Industrial Regulation. This is an important and interesting issue. However, some points need to be clarified or explained before being considered for publication. Below are further elaborations of my viewpoints:

  1. The language need to improve.
  2. In the section of Introduction, a large number of Korean literature were cited in this paper. The authors are encouraged to survey some Asian or international general pictograms. For example, there are some ISO standards on radiation hazards. After all, there may be some foreigners at the scene when a radiation accident occurs.
  3. According to the authors’ description, the meaning and concept of pictograms includes different functions such as guide, order and symbol. The authors are encouraged to further explain which function of the pictograms are discussed in this study.
  4. In the section of Study object and materials collection, 200 respondents are obviously ordinary people living in Jeollabukdo, not professionals or engineers according to the authors’ description. However, the six warning pictograms used to investigate the level of exact recognition were enacted and utilized in accordance with the Korea Industrial Regulation concerning medical and natural radiation safety. The authors are encouraged to explain whether the level of recognition is related to the professional background of the respondent. For example, are there differences for nuclear engineers or bankers? Is there a difference when the respondent is not working in a medical institution?
  5. In the section of Comparison of average, the result shows that male’s mark of right answer is found to be significantly higher than that of female. The authors are encouraged to explain why.
  6. Is it feasible to coexist text and graphics? Is it banned in Korea?

Author Response

Thank you for your review, and Please see a revised manuscript in the attachment.

Comment 1: The language need to improve.

Response:

As per your suggestion, we have re-checked the entire paper and made several changes to improve flow and readability.

Comment 2: In the section of Introduction, a large number of Korean literatures were cited in this paper. The authors are encouraged to survey some Asian or international general pictograms. For example, there are some ISO standards on radiation hazards. After all, there may be some foreigners at the scene when a radiation accident occurs.

Response:

The following international academic papers have been cited in the revised paper as per your suggestion:

Caffaro, F., Cavallo, E. (2015). Comprehension of safety pictograms affixed to agricultural machinery: A survey of users, Journal of Safety Research, 55, 151-158.

Clayton, S. N., Perlotto, C. N. (1997). Comprehension of aviation safety pictograms: Gender and prior safety card reading influences, Processing of the Human Factors and Ergonomics Society Annual Meeting, 2, 806-810.

Jentsch, F. G. (1996). Understanding of aviation safety pictogram among respondents from Europe and the United State, Processing of the Human Factors and Ergonomics Society Annual Meeting, 2, 820-824.

Lim, H. K., Kim, D. H., Ko, B. I. (2000). Cognition of hazard levels with safety signs and pictograms in Korea, Processing of the Human Factors and Ergonomics Society Annual Meeting, 4, 672-675.

Accordingly, the International Organization for Standardization (ISO) has standardized the different pictograms, setting the ISO 7001 (public information symbols) for information pictograms in public facilities and the ISO 7010 (standard for safety symbols) for safety information pictures.

Round 2

Reviewer 1 Report

Thank you for addressing all questions, comments, and suggested revisions.

Reviewer 2 Report

The manuscript have been revised according to my comments. No further comments.